# Sub-ps Laser Deposited Copper Films for Application in RF Guns

**DOI:** 10.3390/ma16031267

**Published:** 2023-02-02

**Authors:** Antonella Lorusso, Zsolt Kovács, Barnabás Gilicze, Sándor Szatmári, Alessio Perrone, Tamás Szörényi

**Affiliations:** 1Department of Mathematics and Physics “E. De Giorgi”, University of Salento and National Institute of Nuclear Physics, 73100 Lecce, Italy; 2Department of High Energy Experimental Particle and Heavy Ion Physics, Wigner Research Centre for Physics, H-1121 Budapest, Hungary; 3Institute of Physics, University of Szeged, H-6720 Szeged, Hungary; 4Department of Optics and Quantum Electronics, University of Szeged, H-6720 Szeged, Hungary

**Keywords:** sub-ps laser ablation, copper thin films, ultrashort pulsed laser deposition

## Abstract

Copper thin films are intended to serve as a cover layer of photocathodes that are deposited by ablating copper targets in a high vacuum by temporally clean 600 fs laser pulses at 248 nm. The extremely forward-peaked plume produced by the ultrashort UV pulses of high-energy contrast ensures fast film growth. The deposition rate, defined as peak thickness per number of pulses, rises from 0.03 to 0.11 nm/pulse with an increasing ablated area while keeping the pulse energy constant. The material distribution over the surface-to-be-coated can also effectively be controlled by tuning the dimensions of the ablated area: surface patterning from airbrush-like to broad strokes is available. The well-adhering films of uniform surface morphology consist of densely packed lentil-like particles of several hundred nm in diameter and several ten nm in height. Task-optimized ultrashort UV laser deposition is thereby an effective approach for the production of thin film patterns of predetermined geometry, serving e.g., as critical parts of photocathodes.

## 1. Introduction

Being the most robust against degradation for its chemical inertness and resistance to high electric breakdown due to its high surface fields, copper-based photocathodes are preferred in S-band radio-frequency guns [1,2,3,4]. Since the quantum efficiency of the order of 10^−5^ at 266 nm laser driven-photoemission of copper, QE_Cu_ [5,6] is relatively low compared to those of other metallic photocathodes, e.g., QE_Y_ = 3 × 10^−4^ and QE_Mg_ = 7.6 × 10^−4^ [7], several solutions were proposed to increase the photoemissive properties of the photocathodes. Wang et al. inserted a Mg disc into the copper base by friction welding [8], while Qian and collaborators did the same by press fitting [9]. These structures suffered from RF breakdown at the Mg–Cu interface [10]. Deposition of a high QE material onto the Cu back plate of the RF gun cavity was another approach [11,12,13]. Among the deposition techniques applied, pulsed laser deposition (PLD), proved to be rather promising since it provided uniform and adherent films, even at room temperature [14,15,16,17,18]. However, fast thinning of the high QE film during frequent laser cleaning, mandatory for removing the surface contaminations to preserve the photoemission performance [19,20], set hard lifetime limits to the photocathodes of this type. To avoid these shortcomings, Lorusso et al. [21] proposed a new photocathode configuration coined “non-conventional”, in which a disc of durable Y or Mg was covered by a copper layer with a central 5 mm diameter emitting area left open. In this structure, the source of photoemission is a robust, long lasting bulk piece of high photoemission efficiency material, while the copper cover layer ensures electrical compatibility with the RF gun made of Cu.

In this paper, results of a feasibility study intended for assessing the potential of sub-ps PLD at 248 nm in producing the copper cover layer of the reversed photocathode structure [21] are reported. Highly efficient localized deposition of well sticking particulate-free films of micrometer thickness is realized by exploiting the unique characteristics of the laser used, with particular regard to the role of the laser spot size on the target surface in controlling the material distribution over the surface to be covered.

In PLD, the habitual control parameter is the energy density. Both probe- and film-based techniques proved that in ns PLD, the plasma broadened with decreasing energy density at fixed spot size. Otherwise, when keeping the energy density constant, the polar distributions became more sharply forward-peaked as the spot size increased. Out of the two parameters, the spot size appeared to be more effective in controlling the distribution [22].

When ablating with ultrashort lasers, the scenario became a bit more complex: while the angular width of the deposits increased with increasing fluence, the ion flux narrowed [23,24,25,26]. To make the picture even more complicated, the angular divergence of the atomic species within the plume was larger than that of the nanoparticles [27]. The effect of changing spot size remained the same: keeping the energy of the ablating pulses constant, the plume geometry deviated from spherical to cylindrical symmetry with increasing spot size [28], leading to narrowing in the angular distribution of the resulting film with a concomitant increase in film thickness [29,30].

Though relevant experiments are relatively scarce, those available e.g., [28,29,30] suggest that in PLD, the size of the ablating laser spot on the target surface controls effectively (yet not independently) both the growth rate and the lateral distribution of the film material. In this report we reveal how the potential of this effect can be exploited for production of thin film patterns of predetermined geometries.

## 2. Materials and Methods

A critical parameter of the ultrashort laser system is the temporal contrast defined as the main pulse–background intensity ratio. In materials processing applications where the pulse energy, rather than the intensity, is the principal parameter, the ratio of the energy of the main fs pulse to the energy of the background, i.e., the energy contrast, is critical. A peculiarity of the Szatmári-type laser system used in this series of experiments is that it has been optimized for high energy contrast by amplifying the frequency converted from 248 nm pulses in a twin-tube amplifier module with a 2-pass amplification geometry in each tube. Thereby producing pulses of 20 mJ maximum energy with an energy contrast, E_fs_/E_ASE_ of 19 [31]. The 600 fs pulse duration was measured by autocorrelation technique using a two-photon ionization based non-linear detector. A 500 mm f/10 Suprasil lens focused the beam at an angle of 45° onto a 99.99% pure copper target of 1” diameter producing an elliptical spot of fairly homogeneous energy distribution on the target surface. The laser impact area was varied by translating the focusing lens along the optical axis while keeping the focal point inside the target. Though using UV grade optical components, the energy loss accounting for the steering optics, focusing lens, the entrance window, and the protective quartz plate behind the window resulted in E_total_ = 4.4 ± 0.2 mJ at the target surface, meaning thereby E_fs_ = 4.18 ± 0.19 mJ and E_ASE_ = 0.22 ± 0.01 mJ. All results reported refer to these fixed pulse energies while changing the geometry of the elliptical spot.

Depositions were performed in a UHV chamber with a base pressure of 10^−4^ Pa at room temperature (Figure 1a). Preliminary experiments revealed that several thousand pulses were necessary and enough to grow films with thicknesses of hundreds of nanometers, thus the number of shots was fixed to 10,000. The normal operational condition of the hybrid dye-excimer laser system used was to produce single pulses with repetition rates up to 10 Hz. In the experiments it was running at 2 Hz. The Si substrates placed parallel to and in front of the target at a distance of 30 mm were covered with copper meshes of 1.00 ± 0.02 mm aperture and 0.25 ± 0.01 mm wire diameter (Figure 1b). Deposition through these masks resulted in an array of columns of ~1.0 × 1.0 mm^2^ cross-sectional areas providing more than 900 height values per sample for the accurate determination of the thickness distribution. The topography of the pattern was measured by a Veeco Dektak-8 stylus profilometer (Bruker Corporation, MA, USA). The surface morphology of the films was characterized by scanning electron microscopy, model JEOL JSM-6480 LV (JEOL Ltd., Tokyo, Japan), and atomic force microscopy, model PSIA XE-100 (Park Systems, Suwon, Republic of Korea) in tapping mode.

## 3. Results

As exemplified in Figure 1, the expanding plasma was extremely forward directed for all focusing conditions, a consequence of the cleanliness of the ablating laser pulse. This jet-like character of the plasma plume was responsible for the striking growth rates and lateral distributions to be reported.

Films were grown at five positions of the focusing lens along the optical axis, resulting in five pairs of spot sizes in the plane of the target surface, since, due to the differences in the directional properties the sub-ps and ns components of the ablating pulse produced different impact areas. The larger areas, referring to the ASE part, extended from 0.30 to 3.30 mm^2^ as measured on highly sensitive UV photopaper. The area ratios were in good agreement with those derived from intensity contrast measurements based on far-field focus diagnostics, underpinning the reliability of the results. Spot areas ranging from 0.085 to 1.01 mm^2^ were derived for the fs component from the analysis of SEM images taken on the processed Cu target surface, as exemplified in Figure 2, again in perfect agreement with the calculated ones.

The deposition rate is routinely quantified as the ratio of maximum thickness to the number of pulses e.g., [29,30]. The dependence of the maximum film thickness on the impact area produced by the sub-ps component of the processing pulse is shown in Figure 3. While the tendency is in line with the expectations [28,29,30], the extent of the effect is intriguing: the thickness steeply rises with increasing area from 0.085 up to 0.34 mm^2^, even with diminishing fluence, followed by an apparent leveling, ending up with 1130 ± 50 nm at 1.01 mm^2^, thus yielding an impressive growth rate of 0.113 nm/pulse here.

The maximum thickness without the lateral distribution of the film material provides not only an incomplete description, but could even be misleading when assessing the effectiveness of film growth. The matrix of columns defined by the mask covered a fairly large area ranging up to ±30° in both directions, constituting a solid base for the derivation of the thickness distributions. As exemplified in Figure 4, for the smallest and largest spot sizes, the changes in the lateral distributions of the film material are equally impressive. Indeed, the increase in maximum thickness with increasing spot area is associated with a striking narrowing in distribution. The results show, within a 10% respect to the maximum thickness value, a lateral angular distribution of around 8° and 25° for 1.01 mm^2^ and 0.085 mm^2^, respectively. The effect is again in line with the observations [28,29,30] and theoretical predictions [22], however, the concentration of the material deposited from the extremely forward-peaked plasma in such close vicinity to the center of symmetry is unique.

Low magnification SEM micrographs revealed that the films consisted of particles of several hundred nanometer diameters distributed homogeneously throughout the whole area (Figure 5). The lateral dimensions of the overwhelming majority of the building blocks constituting the films were no larger than 4–500 nm in diameter.

Dektak records parallel to the major and minor axes of the deposits yielded not only hundreds of height figures, but also a tremendous amount of data on the axial and lateral dimensions of the constituting particles without notable differences either in their size or in the lateral distributions as a function of spot size. AFM images of surface areas exemplified in Figure 6 confirmed the results derived from the SEM analysis and Dektak data: the films were composed of lentil shaped particles of several tens of nanometer in height and approximately one order of magnitude larger in diameter.

Showing typical landscapes after ablation of a material of high thermal diffusivity with sub-ps pulses [32], close-ups of the processed surface exemplified in Figure 7 revealed the origin of the particles: traces of rapid solidification following melting with sharp burrs and small droplets either remained riding on the ridges or redeposited as frozen monuments of the violent ejection of molten material are shown. The overwhelming majority of the droplets are within the 100–500 nm diameter range. In the 0.5–1 µm diameter domain, the number of the globules steeply decreases with increasing diameter and with very few species above one micrometer. This morphology is consistent with rapid expulsion of liquid and vapor droplets which then cooled quickly and resolidified onto the substrate with partial redeposition, giving a clue about the morphology of the films presented in the preceding paragraph.

## 4. Discussion

The camera snapshots (Figure 1) clearly documented the extremely forward directed appearance of the plume. It is worth noting that while results had rather different thickness distributions depending on the actual spot size, the visual appearance of the plume remained apparently unchanged. Though observation of such a high plume length/width ratio is exceptional indeed, for initial plasma thicknesses of tens of nanometers with millimeter lateral dimensions, the theoretical considerations admit even higher ratios [33].

To reveal what is behind this behavior, let’s first assess the contributions of the fs vs. ns components of the processing pulse to the process. Since the energies of the ASE and the main pulse and the respective spot sizes on the target surface are different, the effects of two fluence (and intensity) series should be examined. Table 1 displays the process parameter window of the experiments in terms of fluence and intensity, calculated separately for the 600 fs main pulse of 4.18 ± 0.19 mJ energy and the ASE background conveying 0.22 ± 0.01 mJ energy in 15 ns. Since the main pulse lies in the middle of the background, the ASE contribution to the ablation was calculated with 0.11 mJ and 7.5 ns.

The ns ablation thresholds of copper range from 0.3 J/cm^2^ [34] through 0.5 J/cm^2^ [35] and 2.0–2.5 J/cm^2^ [36] up to 5–6 J/cm^2^ [35,37,38]. In the context of this study the most relevant figure is 3.4 ± 0.5 J/cm^2^ reported for KrF excimer laser processing [39,40]. Since, in our case, the highest ASE fluence value of 0.037 J/cm^2^ calculated for 0.30 mm^2^ spot size was two orders of magnitude lower than the most relevant figure [39,40] and even the absolute lowest one of 0.3 J/cm^2^ reported [34] is one order of magnitude higher, it is safe to say that in our experiments the ns component did not ablate. The intensity figures in Table 1 further strengthen this statement: since below a prepulse level of 10^7^ W/cm^2^, the amplified spontaneous emission of the amplifier did not cause measurable preplasma formation or photo-evaporation of the target material [41], the 4.9 × 10^6^ maximum ASE intensity was definitely not enough for producing plasma components. In conclusion: due to the high-energy contrast ratio, a unicity of the laser system used, in our case the ns pedestal did not contribute to the ablation and had no impact on the fs part either, consequently all effects obtained should be assigned exclusively to the sub-ps main pulse.

For pulses shorter than ~1 ps, the melting depth proved to be independent of pulse duration for copper [42]. Therefore, the excellent agreement in the ablation thresholds reported at F_th_ = 0.50 ± 0.05 J/cm^2^ [42,43,44,45,46] is reasonable. Nevertheless, material removal at extremely low rates, slowly increasing with fluence was shown to be possible even below 0.5 J/cm^2^ [47,48,49], while 0.35 and 0.15 J/cm^2^ were derived as thresholds in 500 fs@248 nm PLD [40,50].

In the light of these figures, it is apparent that all fluence values derived for the main pulse (Table 1) were above threshold. Lying in the high-fluence regime, 4.92 J/cm^2^ should have produced high ablation/deposition rate, while approaching the threshold, 0.41 J/cm^2^ should have been much less effective. The intriguing effect is that the rise in the spot areas from 0.085 to 1.01 mm^2^ not only compensated for the concomitant decrease in fluence, but even resulted in a continuous rise in maximum film thickness. Certainly, the concentration of the material in close vicinity of the center of symmetry as a result of the narrowing angular distribution of the ejecta also contributed to the evolution of the maximum thickness with increasing spot size. Nevertheless, despite the fact that the decreasing removal rate, due to the vanishing fluence with increasing spot area, the total amount of material deposited also proved to be apparently higher for 1.01 mm^2^ (Figure 4.) This revealed that the size of the ablating laser spot on the target surface more effectively controlled both the growth rate and the lateral distribution of the film material than the fluence. From a practical viewpoint: the control of the material distribution by tuning the spot area allows for optimalization of the deposition process for the actual application by writing patterns through moving the plume on the surface to be covered like an airbrush using large spot sizes or painting with broad strokes produced by small spots.

In order to judge the efficiency of our approach, the growth rates derived were compared to those reported in [40], actually one of the first studies performed using a 500 fs@248 nm laser. In this study, focusing the beam onto a 0.05 mm^2^ spot resulted in growth rates between 2 × 10^−3^ and 3 × 10^−3^ nm/pulse within the 0.4–6.0 J/cm^2^ fluence domain. Since at the same fluence, the amount of material ablated scales with the spot area [34] the figures given in [40] have been recalculated for the spot areas referring to the main pulse in Table 1. The growth rates obtained this way and those calculated from Figure 3 are compared in Figure 8.

The similarity of the two plots reveals that the spot size dependence is real and characteristic of the lasers of this class. The 3–4 times higher growth rates of magnitude 0.1 nm/pulse reported in our case are impressive even in comparison with the literature’s data [40,49,51]. What makes a difference is that the early version of the laser system [40] definitely possessed lower energy contrast due to the single-pass on-axis amplification scheme. The high growth rates produced by ablating with clean pulses highlight the decisive role of the energy contrast in determining the efficiency of the process.

In the process parameter domain covered in this study, the ablated material is predominately decomposed as nanoparticles, with atoms and ions as minor components e.g., [24,52]. As evidenced by SEM/AFM mapping, the morphology of the films resembles the surface of copper films deposited by pulses of 5 ps duration@248 nm [49]. The reason for the presence of relatively large sized particles in the films is their origin: they are ejected directly from the molten target surface [47].

Excellent sticking is a characteristic feature of the unusually high forward peaking. When forward peaking is increased, the amount of energy converted from thermal energy to directed plume expansion kinetic energy increases [53]. The increasing electron density leads to increased pressure gradients at the plume edge, resulting in increased flow speeds. High forward peaking also means high ion density. Even around 10^13^ W/cm^2^, the ions arrive at the substrate surface with velocities in the range of 10^6^–10^7^ cm/s [54,55]. The high-energy ions penetrate into the substrate several atomic layers deep, practically realizing ion implantation, anchoring the film to the substrate [56]. Excellent sticking even with the maximum thickness of 1.2 μm to the flat Si wafer surface was verified, indeed: the films not only passed the “Scotch tape’’ test but could hardly be removed even by sandpaper.

The extremely forward-peaked plume produced by the ultrashort UV pulses of high-energy contrast ensures fast film growth of thickness higher than 1 μm obtained in a reasonable time of 1 h and 30 min, with the highest spot area of 1.01 mm^2^, and a deposition rate of 0.11 nm/pulse. Achieving such a high adherent film within a short deposition time makes PLD induced by sub-ps laser very interesting for developing metallic photocathodes that are useful in RF guns. In these guns, operating at the S-band (2.856 GHz), the features of photocathodes based on thin films are of a thickness comparable with the copper skin depth of 1.2 μm and, at the same time, the appropriate morphological and structural characteristics to face the harsh working conditions in an RF environment.

## 5. Conclusions

The results presented demonstrate that in an ultrashort temporal regime, the tuning of the ablating spot area offers a broad dynamic range in controlling both the maximum thickness and the lateral distribution of the film material. Ablation with pulses of high energy contrast composed of a 600 fs component of (4.18 ± 0.19) mJ energy and an ASE background conveying (0.22 ± 0.01) mJ energy in 15 ns produces excellently sticking films with a maximum thickness exceeding 1.1 μm for 1.01 mm^2^ laser spot area with a very forward-peaked shape, while a quite uniform lateral distribution was found for an 0.085 mm^2^ laser spot area with a maximum thickness of 300 nm. This study was important for understanding that PLD with clean UV pulses is thereby a versatile effective technique for growing metallic thin films with appropriate characteristics to be used as photocathodes for application in RF guns operating at the S-band.

## Figures and Tables

**Figure 1 materials-16-01267-f001:**
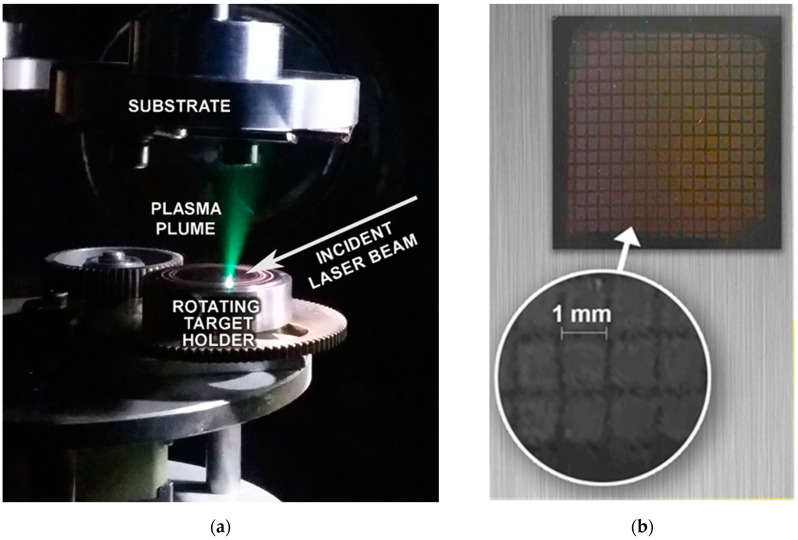
(**a**) The plasma plume expansion during ablation; (**b**) the mesh used for defining the array of patterns deposited.

**Figure 2 materials-16-01267-f002:**
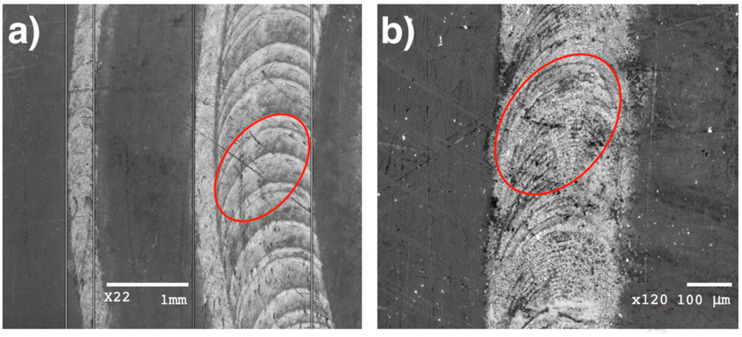
Low magnification SEM images of tracks written by tight and loose focusing with: (**a**) marking the contour of a spot of 1.01 mm^2^ area and (**b**) a close-up of the track written focusing to the smallest spot of 0.085 mm^2^.

**Figure 3 materials-16-01267-f003:**
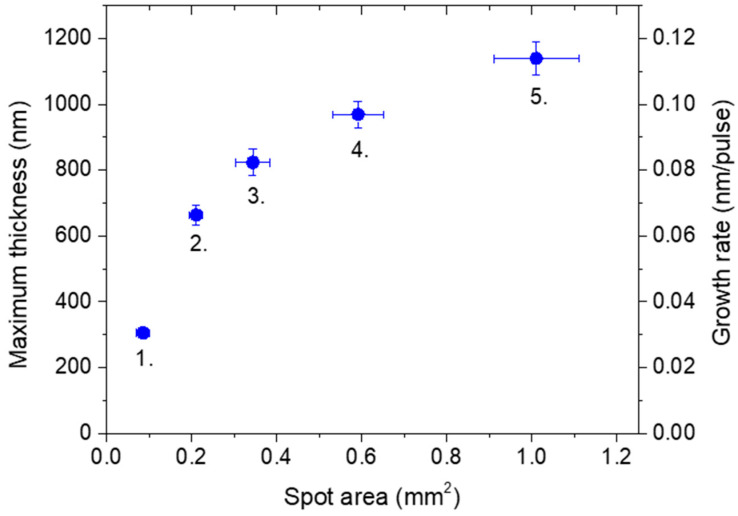
The evolution of maximum thickness and the growth rate derived with rising spot area. The numbers 1.–5. refer to lens positions to facilitate the assignment of the thickness data to the respective process parameters listed in Table 1.

**Figure 4 materials-16-01267-f004:**
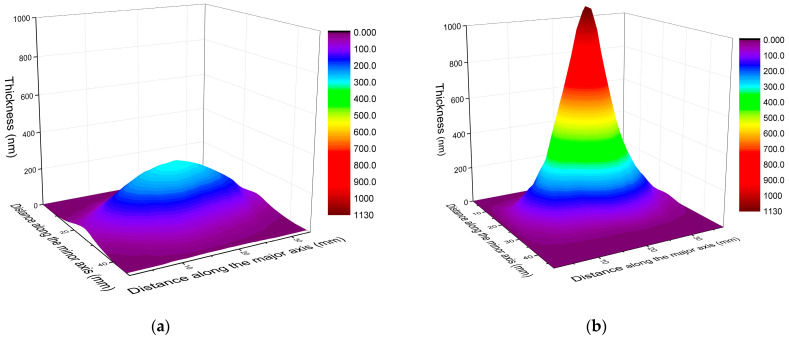
The thickness distribution of films deposited at: (**a**) 0.085 mm^2^ and (**b**) at 1.01 mm^2^ spot areas. The thickness scales have been set identical intentionally to highlight the difference.

**Figure 5 materials-16-01267-f005:**
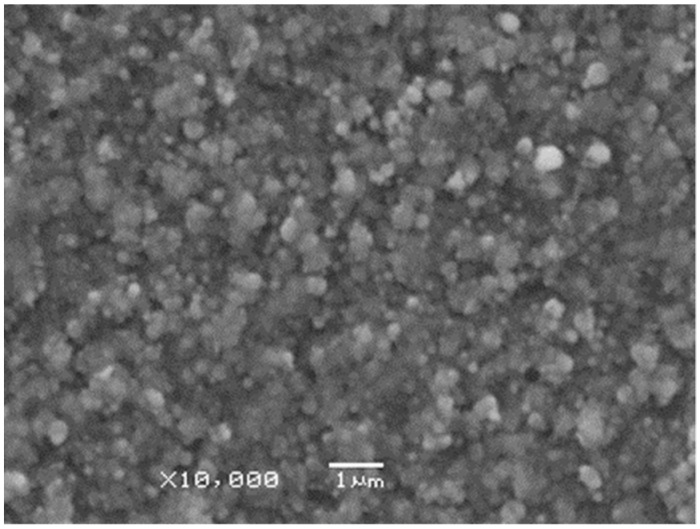
A low magnification SEM micrograph taken on a Cu film deposited by 10,000 pulses focused onto 0.085 mm^2^ area on the target surface.

**Figure 6 materials-16-01267-f006:**
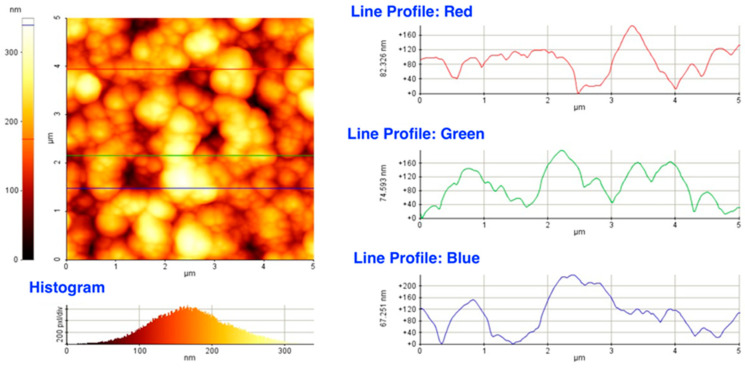
An AFM image and line profiles revealing the axial dimensions of the building blocks of the films.

**Figure 7 materials-16-01267-f007:**
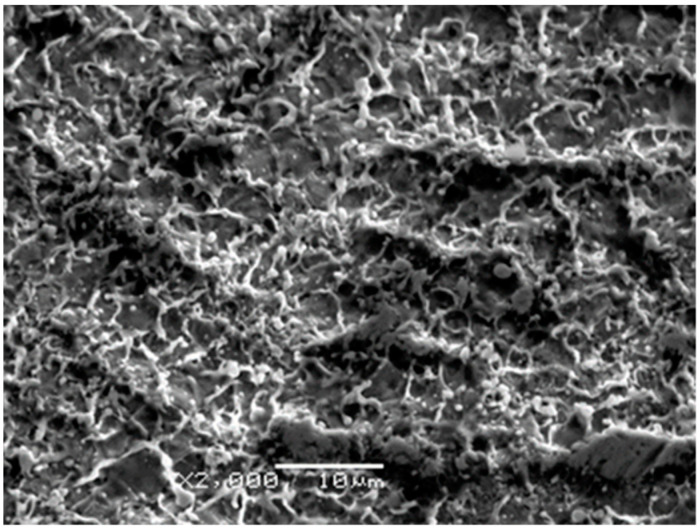
SEM image taken on a segment of a track ablated by 10,000 pulses focused onto 0.085 mm^2^ area on the target surface.

**Figure 8 materials-16-01267-f008:**
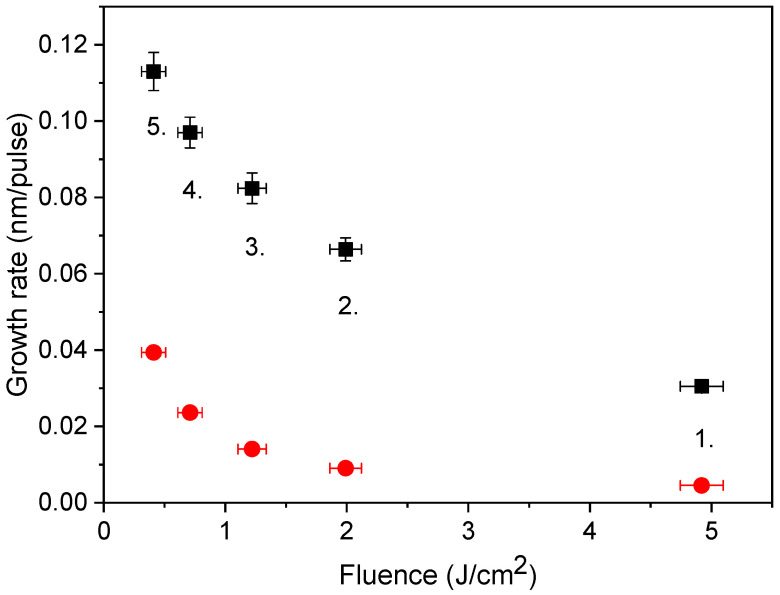
Comparison of growth rates of copper films deposited by sub-ps UV pulses: this work (■) and data points from [40] converted for the same spot sizes (●).

**Table 1 materials-16-01267-t001:** The process parameter windows.

Lens Position		1	2	3	4	5
Main Pulse	Area (mm^2^)	0.085	0.21	0.34	0.59	1.01
	Fluence (J/cm^2^)	4.92	1.99	1.22	0.71	0.41
	Intensity (W/cm^2^)	8.2 × 10^12^	3.3 × 10^12^	2.0 × 10^12^	1.2× 10^12^	0.7 × 10^12^
ASE	Area (mm^2^)	0.30	0.90	1.48	2.16	3.30
	Fluence (J/cm^2^)	0.037	0.012	0.0074	0.0051	0.0033
	Intensity (W/cm^2^)	4.9 × 10^6^	1.6 × 10^6^	9.9 × 10^6^	6.8 × 10^6^	4.4 × 10^6^

## Data Availability

The data presented in this study are available on request from Tamás Szörényi, t.szorenyi@physx.u-szeged.hu.

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
