# Peer review of "Sub-ps Laser Deposited Copper Films for Application in RF Guns"

_materials, 2023, doi:10.3390/ma16031267_

Round 1
Reviewer 1 Report
The manuscript presents pulsed laser deposition of copper films on a substrate with UV sub-picosecond pulses. The experiments are rigorous and the results are quite interesting. I have a couple of comments authors may want to address.
1. The film growth rate is decreasing with the fluence as shown in Fig. 8. Is the plasma shielding (with an increase in no. of pulses) influencing the material removal rate and this in turn affecting the film growth rate? Please comment.
2. What could be the effect of applying burst mode (splitting a single pulse into a train of sub-pulses) as this was shown to increase the material removal rates of copper without inducing too much heat? Please comment.
Reviewer 2 Report
Lorusso et al. presents work that deposition of copper film using sub-ps laser. There are some comments for the authors.
Major comments:
1. Since the parameters of the laser is important, how did the authors measure the pulse width?
2. How did the authors achieve 2 Hz pulse? What is the original repetition rate of the laser? Can you use higher repetition rate? This should be discussed in the manuscript.
Minor comments:
1. Line 35, 36,134, 136, etc. please correct all the superscription.
2. The abbreviation of the terminology should be in a parentheses after the first use of the word. Like “pulsed laser deposition, PLD” should be “pulsed laser deposition (PLD)”. Please correct it in the manuscript.
